METHODS AND RESOURCES

# Subcellular proteomics of the protist *Paradiplonema papillatum* reveals the digestive capacity of the cell membrane and the plasticity of peroxisomes across euglenozoans

Michael J. Hammond[1,2,3*], Orsola Iorillo[1,2], Drahomíra Faktorová[1,2], Michaela Svobodová[1,2], Bungo Akiyoshi[4], Tim Licknack[3], Yu-Ping Poh[3], Julius Lukeš[1,2], Jeremy G. Wideman[3*]

**1** Institute of Parasitology, Biology Centre, Czech Academy of Sciences, České Budějovce (Budweis), Czech Republic, **2** Faculty of Science, University of South Bohemia, České Budějovce (Budweis), Czech Republic, **3** Center for Mechanisms of Evolution, Biodesign Institute, School of Life Sciences, Arizona State University, Tempe, Arizona, United States of America, **4** Institute of Cell Biology, School of Biological Sciences, University of Edinburgh, Edinburgh, United Kingdom

\* michael.hammond@paru.cas.cz (MJH); jeremy.wideman@asu.edu (JGW)

## Abstract

Diplonemids are among the most diverse and abundant protists in the deep ocean, have extremely complex and ancient cellular systems, and exhibit unique metabolic capacities. Despite this, we know very little about this major group of eukaryotes. To establish a model organism for comprehensive investigation, we performed subcellular proteomics on *Paradiplonema papillatum* and localized 4,870 proteins to 22 cellular compartments. We additionally confirmed the predicted location of several proteins by epitope tagging and fluorescence microscopy. To probe the metabolic capacities of *P. papillatum*, we explored the proteins predicted to the cell membrane compartment in our subcellular proteomics dataset. Our data revealed an accumulation of many carbohydrate-degrading enzymes (CDZymes). Our predictions suggest that these CDZymes are exposed to the extracellular space, supporting proposals that diplonemids may specialize in breaking down carbohydrates in plant and algal cell walls. Further exploration of carbohydrate metabolism revealed an evolutionary divergence in the function of glycosomes (modified peroxisomes) in diplonemids versus kinetoplastids. Our subcellular proteome provides a resource for future investigations into the unique cell biology of diplonemids.

## Introduction

Diplonemids are unicellular, heterotrophic eukaryotes, which constitute one of the most abundant and species-rich protist groups within the world's oceans [1,2]. In addition, recent investigations show a comprehensive distribution of diplonemids in

which permits unrestricted use, distribution, and reproduction in any medium, provided the original author and source are credited.

**Data availability statement:** Data are available via ProteomeXchange with identifier PXD065121.

**Funding:** This work was supported by the National Science Foundation BII: Mechanisms of Cellular Evolution DBI-2119963 awarded to J.G.W., the Czech Grant Agency grants 23-06479X and 25-15298S awarded to J.L. and the Wellcome Discovery Award 227243/Z/23/Z awarded to B.A. The funders had no role in study design, data collection and analysis, decision to publish, or preparation of the manuscript.

**Competing interests:** The authors have declared that no competing interests exist.

**Abbreviations:** AA, amino acid; CDZymes, carbohydrate-degrading enzymes; ER, endoplasmic reticulum; FBP, fructose 1,6-biphosphatase; GADPH, glyceraldehyde 3-phosphate dehydrogenase; GPI, glycosyl-phosphatidylinositol; KM, K-means; LFQ, label-free quantification; PFK, phosphofructokinase; PGAM, phosphoglycerate mutase; PGK, phosphoglycerate kinase; PPP, pentose phosphate pathway; PTS, peroxisomal targeting signals; SVM, support vector machine; TCA, tricarboxylic acid; TMD, transmembrane domains; t-SNE, t-distributed Stochastic Neighbor Embedding.

freshwater environments [3], as well as in all pelagic zones of the ocean [4,5]. Global metabarcoding estimates >67,000 species of diplonemids worldwide, and therefore, they are presumed to be key ecological players in all marine ecosystems [6].

Despite their importance, our knowledge of diplonemid nutrition strategies, ecological roles as well as their molecular and cellular biology remains limited. Beyond general heterotrophy [7], investigating their lifestyles and specific feeding modes remains challenging, partly due to the difficulty in observing diplonemid behavior in nature. By contrast, the relative ease by which diplonemids can be established in stable axenic cultures (typically in protein-rich media) is promising, and makes them amenable to an expanding range of genomic, transcriptomic, and proteomic experiments [8,9]. Such techniques are necessary to further characterize diplonemids' cellular and ecological functions.

A high-quality nuclear genome is available for the diplonemid *Paradiplonema papillatum* (formerly *Diplonema*) [9], with two recent assemblies now available for *Diplonema japonicum* [10] and *Rhynchopus euleeides* [11], in addition to several previously existing transcriptomes [6]. However, *P. papillatum* remains the only genetically tractable diplonemid, enabling functional investigations by gene deletion [12], endogenous tagging of proteins [13], and immunoprecipitation [14]. Such tractability has allowed the investigation of *P. papillatum* respiratory complexes [15], mitochondrial ribosomes [14], and kinetochores [16]. Diplonemids retain many genes that can be traced to the last eukaryotic common ancestor, including rare, restricted homologs referred to as jotnarlogs [17]. Thus, diplonemids may prove particularly informative for understanding the complexities of the ancestral eukaryote [18].

Among the many protein-coding genes predicted from its genome, an unexpected finding in *P. papillatum* was the identification of several hundred carbohydrate-degrading enzymes (CDZymes), with the capacity to digest pectin, cellulose, and β-1,3 glycans among other carbohydrates [9]. This expanded CDZyme repertoire is particularly prominent compared to their relatives, Euglenida and Kinetoplastea [9]. Such presence implies a proclivity of *P. papillatum* (and potentially other diplonemids) towards digestion of cell wall components of plants and algae. However, it is unclear how these organisms can specifically digest the cell walls of photosynthetic eukaryotes. Osmotrophy has been proposed [7], through secreting enzymes to their exterior, as well as phagotrophy, internally ingesting components for subsequent processing. Though *P. papillatum* is a tractable species, tagging and visualizing hundreds of CDZymes to determine their localization is unrealistic. We therefore sought to perform subcellular proteomics to localize CDZymes to various intracellular compartments.

Here, we use a subcellular proteomics workflow similar to localization of organelle proteins by isotope tagging *via* differential ultracentrifugation (LOPIT-DC) [19], to produce the first subcellular proteome of a diplonemid. With our data, we classified 4,870 proteins to 22 cellular compartments in *P. papillatum*. We validated several predicted locations by epitope and fluorescent tagging. Our subcellular proteome provided a clearly resolved cluster of cell membrane proteins enriched with secreted CDZymes. We suggest these enzymes can actively degrade plant and algal cell walls, initially at the cell's exterior. We also show an ability for internal carbohydrate processing with various secreted CDZymes distributed to the lysosomal compartments, and expand

on traditional carbohydrate metabolism across glycosomes and the cytoplasm, demonstrating their diverged compartmentalization from their sister clade Kinetoplastea [20]. Finally, we reveal an extensive mitochondrial capacity for varied amino acid (AA) digestion, foregrounding the metabolic versatility of this model diplonemid. Our localization of thousands of *P. papillatum* proteins provides a repository of information that will extend our knowledge of diplonemids, facilitating an exploration of their unusual cell biology and function.

## Results and discussion

### Subcellular proteomics allows predictive clustering of *P. papillatum* proteins into 22 distinct compartments

To obtain a subcellular map of *P. papillatum*, we used a modified workflow adapted from a LOPIT-DC protocol described previously [19]. Briefly, cells were grown axenically in "Diplo" media (seawater supplemented with 10% Fetal Bovine Serum and 1 g tryptone). Approximately $9.9 \times 10^8$ cells per sample were collected and lysed in detergent-free lysis buffer in a nitrogen cavitator (250 psi for 10 min). Cell lysates underwent differential centrifugation resulting in 11 distinct fractions, including initial unlysed cells. We used western blot analysis using antibodies against ATP synthase subunit β from *Trypanosoma brucei* [21], mammalian Grp75 [22] and Grp78 [23] to ensure fractional proteomic profiles were distinct (S1 Fig).

Label-free quantification (LFQ) analysis was followed by peptide data analysis in ProteomeDiscoverer [24] and with R, primarily *via* the pRoloc package [25]. Data was quantified against the nuclear and mitochondrial genomes of *P. papillatum* [9]. After quality control, 4,870 unique proteins were detected in this dataset. Following normalization, proteins lacking peptide coverage in all fractions underwent imputation *via* "neighbor averaging" (1,285 proteins) as well as "zero" methods (2,073 proteins, Table A in S1 Data).

To predict cellular localization for the *P. papillatum* subcellular proteome, we manually curated a set of 368 proteins constituting markers with canonical localizations (e.g., mitochondrion, flagellum, cytosol), specific functions (e.g., membrane trafficking compartments) or those with inferred localization data, corresponding to a total of 22 distinct subcellular compartments or protein complexes (Table B in S1 Data). Using a median svm cutoff (Table C in S1 Data), we predicted sub-localization of 2,435 proteins (Fig 1A and 1B), with the remainder additionally classified to these compartments with lower confidence (Table D in S1 Data; S2 Fig). To further corroborate our designated clusters, we mapped predicted target signals and protein features onto the t-distributed Stochastic Neighbor Embedding (t-SNE) distributions (Fig 1C). Mitochondrial target peptides (mTP, predicted *via* TargetP2.0) [26] are abundant across the three mitochondrial clusters—matrix, protein complexes and membrane-enriched. Signal peptides, predicted *via* SignalP6.0 [27] show enrichment across soluble lysosome, cell membrane, endoplasmic reticulum (ER)/Golgi clusters, as well as endocytic and multivesicular membrane trafficking compartments. Finally, transmembrane domains (TMD), predicted via DeepTMHMM [28] correlate to the various membrane-enriched clusters of the diplonemid cell.

Next, we highlighted proteins that exhibit differences in abundance when *P. papillatum* was grown in different media: "Diplo" versus "Hemi" media (seawater supplemented with 10 ml inactivated horse serum and 1 ml/L LB medium), and oxygen-abundant *versus* depleted conditions (Fig 1D) [8]. Cells grown in nutrient-rich "Diplo" medium show enrichment for proteins predicted to the cytosolic ribosome and cell membrane clusters, including sodium/potassium exchangers and sterol transporters. The nutrient-poorer "Hemi" medium showed notable enrichment across multiple clusters, including the proteasome, cytosol, soluble lysosome, and mitochondrial regions (Fig 1D). Equally, aerobic conditions resulted in the enrichment of several hypothetical cell membrane components, subunits of mitochondrial complex IV, as well as various soluble lysosomal proteases. By contrast, anaerobic conditions induced enrichment across clusters of the cytosol, cytosolic ribosomes, mitochondrial matrix, and translation initiation factors 2 and 3 (Fig 1D).

### Endogenous tagging confirms subcellular localizations inferred from proteomic data

To validate designated clusters, we successfully performed endogenous tagging with either V5 or YFP epitopes on 12 proteins predicted or classified to various cell compartments, which typically lack both annotation and homologs outside

PLOS Biology

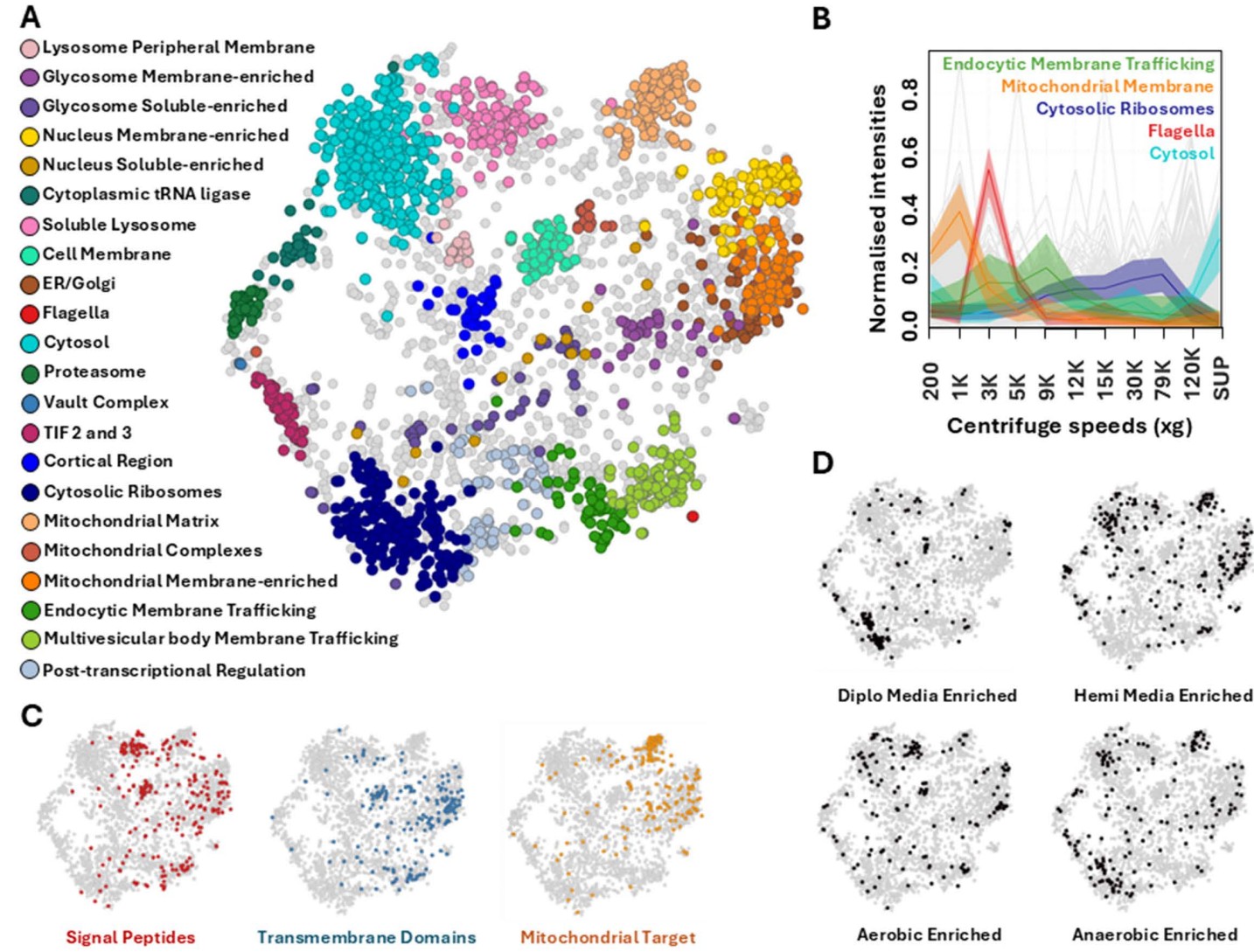

**Fig 1. Clustered protein predictions of *Paradiplonema papillatum* align with predicted protein features and clarify conditional enrichment trends. (A)** Neighbor-average imputed t-SNE of dataset displaying clustered predictions displayed for 2,797 proteins across 22 cell compartments. Predictions were generated via support vector modeling conducted on fractional profiles of marker proteins, applied to the remaining dataset. **(B)** Selected fractional abundances of marker proteins across one replicate of this experiment, representing distinct profiles that facilitate predictive clustering (SUP, Supernatant). **(C)** Software prediction for protein features of signal peptides, transmembrane domains, and mitochondrial target peptides across dataset, demonstrating accumulation across certain defined compartments. **(D)** Proteins determined to be enriched in varying nutrient media (Diplo or Hemi) or cultivation conditions (aerobic or anaerobic) from a conditional study of *P. papillatum* [8]. Additional information for all proteins available in Tables B, C and D in S1 Data.

diplonemids (Fig 2). Such proteins were ultimately located to the flagella (Fig 2A), cytoplasm (B,C), mitochondrion (D,E), ER/Golgi (F), nucleus (G,H,I), nucleolus (J), endocytic membrane trafficking (K), and, finally, the cell membrane (L), encompassing nine defined clusters in total (Table E in S1 Data).

Mitochondrial proteins DIPPA_24150 and DIPPA_15120 co-localize with the organellar DNA within this reticulated mitochondrion (S3 Fig) at the cell periphery (Fig 2D and 2E). In turn, the tagged ER/Golgi candidate DIPPA_04811 shows a signal surrounding the nuclear DNA, while also branching and extending into the cell posterior (Fig 2F). Next, we validated

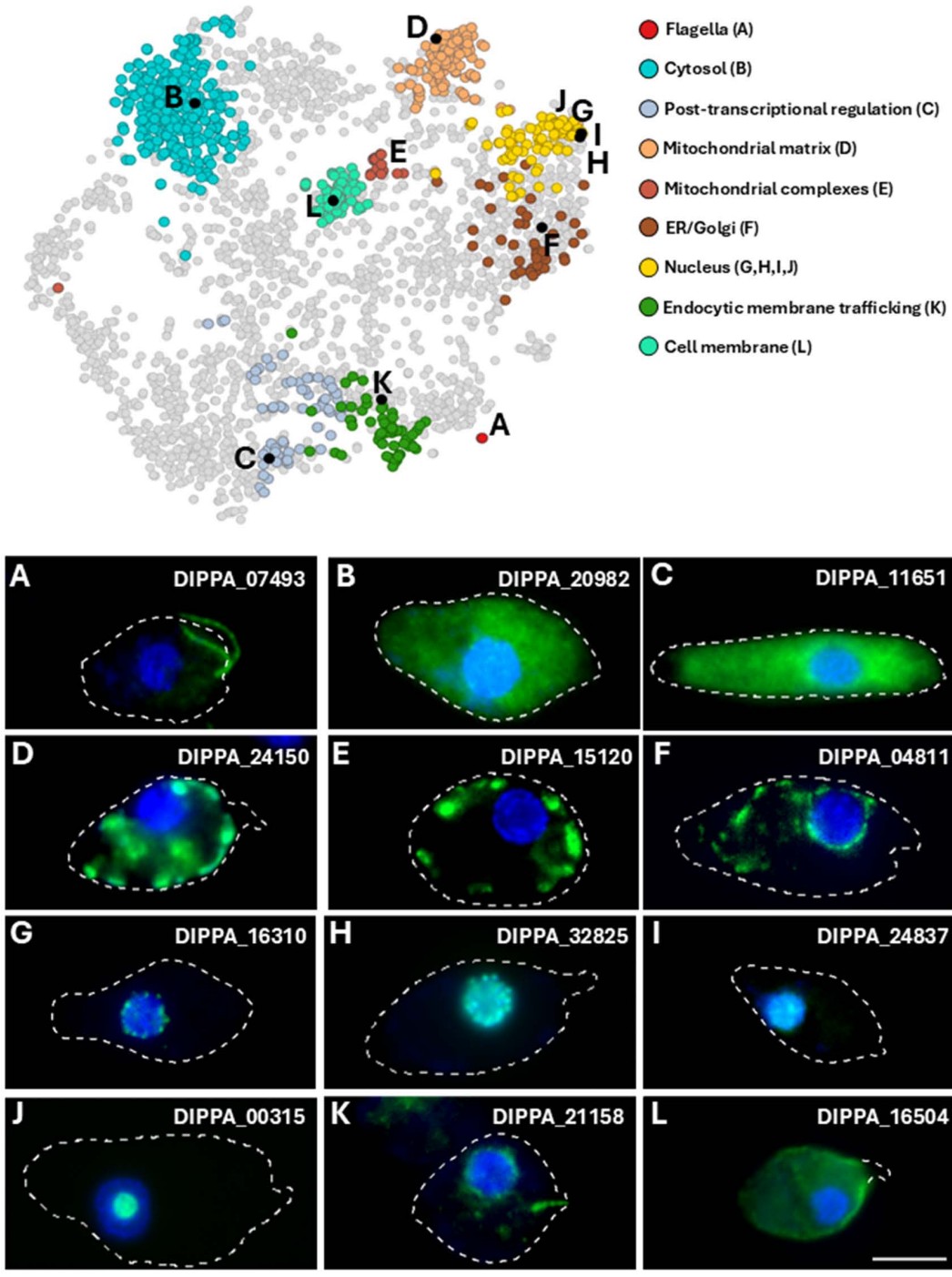

**Fig 2. Endogenous tagging of novel proteins confirms supervised cluster predictions.** Tagged proteins highlighted (black) among relevant predicted clusters, resolved on neighbor-averaged imputed t-SNE. Individual cell lines were generated via endogenous tagging and imaged through fluorescence microscopy for comparison with the compartment-relevant protein was predicted to. Merged microscopy images showing protein signal (green) merged with nuclear and mitochondrial DNA (blue). All imaged cells are oriented with their apical regions facing right and posterior facing left, cell membrane outlines are traced for all images except for **L,** showing only trace of the papilla, which lacks signal. Scale bar represents 5 μm. Proteins **A, E, G,** and **J** are resolved in zero and neighbor-averaged imputed t-SNE in S3 Fig, for which separate channels of each cell line are also shown. Further information on cell lines is available in Table E in S1 Data.

four proteins assigned to the nucleus, which show different sub-localizations by immunofluorescence analysis (IFA) within this compartment (Fig 2G–2J). The first nuclear candidate (DIPPA_16310) has a patchy distribution on the outer-most periphery of the nuclear DNA (Fig 2G). Unlike the novel ER/Golgi protein (Fig 2F), this candidate does not extend beyond the nucleus, and hence, likely constitutes a novel nuclear membrane component. A second nuclear candidate (DIPPA_32825) co-localizes with the chromatin signal of the nucleus (Fig 2H), similar to the general nuclear signal of a third selected nuclear protein (DIPPA_24937) (Fig 2I). The last nuclear candidate (DIPPA_00315) displays a confined distribution within the nucleus, corresponding to the nucleolus (Fig 2J). Similarly, this protein's uncharacterized homolog within the kinetoplastid *T. brucei* (Tb927.3.2750) also displays a nucleolus-like signal when tagged *via* green fluorescent protein [29].

One protein (DIPPA_21158) classified to the 'endocytic membrane trafficking' compartment, seemingly exhibits dual localization, with an ER-like pattern similar to DIPPA_04811 (Fig 2F), while also showing enrichment towards and encompassing the cell cytopharynx (Fig 2K). Finally, a protein predicted to the cell membrane cluster (DIPPA_16504) (Fig 2), shows a signal enriched across the cell outline, excepting the apical papilla (Fig 2L). This protein possesses a signal peptide and a TMD, both of which are enriched for proteins predicted to the cell membrane (Fig 1C). This cell membrane cluster also exhibits an accumulation of predicted signal peptides in tandem with glycosylphosphatidylinositol (GPI)-attachment domains (S4 Fig), further supporting the validity of this newly defined cluster.

## Secreted CDZymes localize to the cell membrane and lysosomes

CDZymes are particularly abundant in *P. papillatum*, suggesting complex digestive capabilities against plant and algal cell wall carbohydrates [9]. Through our subcellular dataset, we show a notable proportion of CDZymes enriched with signal peptides localized with high confidence to the cell membrane and the lysosome (Fig 3). Schematic diagrams of these cell membrane CDZymes show the presence of a C-terminal TMD and/or GPI anchor sites, preceded by the catalytic domains of associated enzymes. This topology indicates that the CDZyme domains are exposed to extracellular space and thus expected to digest external carbohydrate substrates (Fig 3A). Enzymatic domains present include pectin esterase, pectin lyase, and glycosyl hydrolases, from which we construct a digestion pathway on the cell membrane to externally degrade methylated pectin to galacturonic acid monomers (Fig 3A). We note that several proteins of pectin degradation (DIPPA_26123, DIPPA_17995, DIPPA_19114) possess multiple complete enzymatic domains of the same category following their signal peptide (Fig 3A). Internal repeat regions of high similarity in both proteins likely arose from genetic duplication across these regions, with the further presence of repeated parallel beta helix regions within these enzyme domains suggesting a coiled tertiary structure for both of these extracellular-exposed, embedded proteins. Some CDZymes of the cell membrane lack the predicted TMD or GPI anchors, such as glycosyl hydrolase 74 (DIPPA_10258), which degrades hemicellulose to glucose, xylose, and galactose. It remains a possibility that such CDZymes are released into the extracellular space or simply lack identifiable motifs for cell anchorage.

Candidate sugar transporters, recently identified through genome analysis [9], were not localized to the cell membrane cluster, rather being assigned to the ER/Golgi and glycosome compartments (Table D in S1 Data). Thus, we propose that instead of being passaged directly to the cytoplasm across the cell membrane, digested or partially digested carbohydrate substrates are engulfed through the cytopharynx, leading to trafficking through the endocytic vesicles, which have been observed prominently budding off from this distinctive structure in diplonemids [30]. Within the endocytic membrane trafficking cluster of this dataset, we also identified one secretory CDZyme (Fig 3B). Endocytosed contents are typically passaged to the lysosomal compartments, for which we also define a corresponding cluster of soluble proteins containing numerous signal peptide-bearing CDZymes, with the ability to digest various forms of pectin and other polysaccharide chains, such as sucrose and glycosides (Fig 3C). We additionally predict one sugar transporter (DIPPA_16016.mRNA.1) to the multivesicular membrane trafficking body, enriched for V-type ATPases and other membranous components of the lysosome, suggesting eventual saccharide transport from these organelles to the cytosol and possibly other compartments.

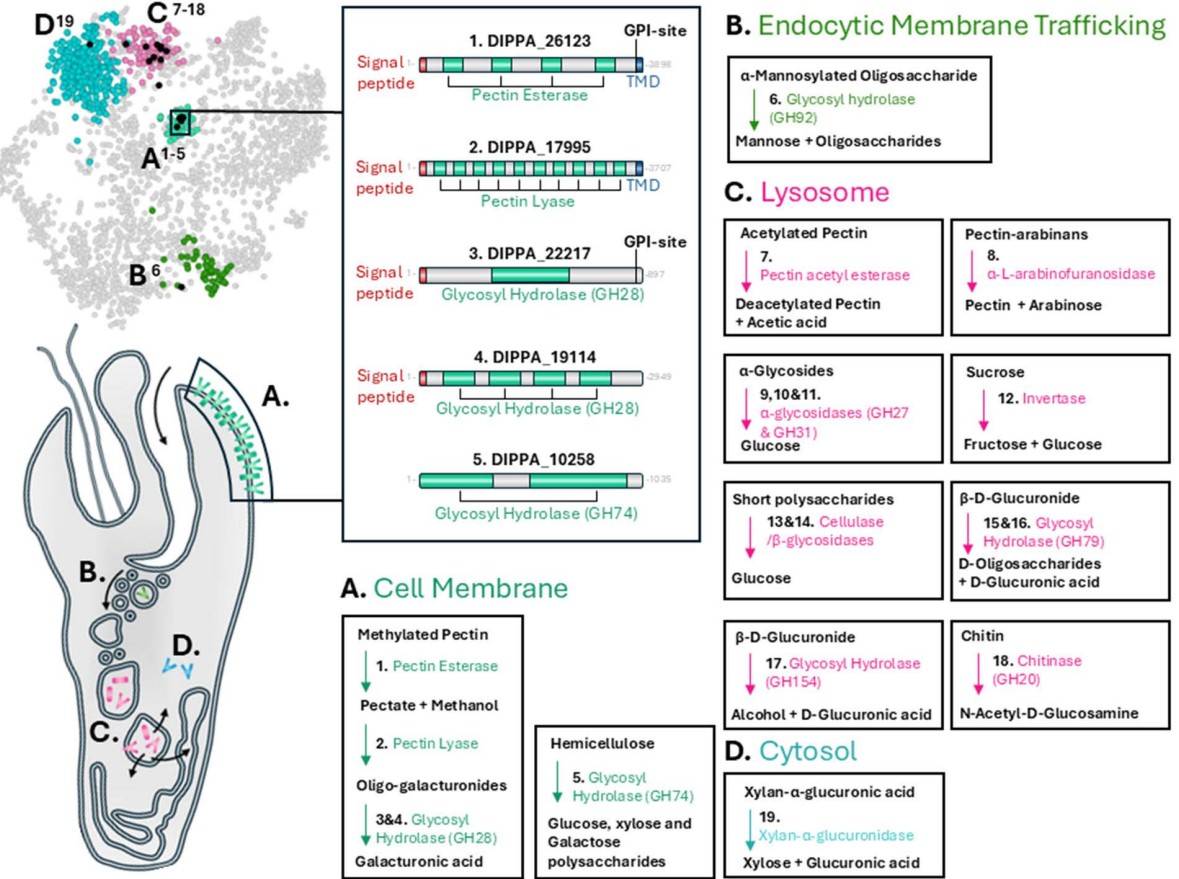

**Fig 3. Secreted carbohydrate-degrading enzymes (CDZymes) primarily localized on cell membrane and lysosomes.** Distribution of signal peptide-enriched CDZymes, which are predicted with high confidence on neighbor-average imputed t-SNE, corresponding to highlighted cluster predictions of the cell membrane **(A)**, endocytic membrane trafficking **(B)**, lysosome **(C)**, and cytosol **(D)**. Proteins of cell membrane **(A)** have schematic representations showing software predictions for signal peptides, transmembrane domains (TMD), and/or GPI attachment sites, which demonstrate extracellular exposure of CDZyme domains in accordance with conventional membrane topology. Bordered outlines indicate separate enzymatic reactions for CDZymes, with carbohydrate substrates and products in black. Further information on CDZymes of *Paradiplonema papillatum* is available in Table F in S1 Data.

Given that these analyzed cells were grown in the protein-rich "Diplo" medium, we did not necessarily expect an abundance of CDZymes in our extractions. Nonetheless, we detected a total of 94 different enzymes across our subcellular dataset, 55 of which were not recorded in previous studies (Table F in S1 Data) [8,9]. The proteomic presence of these enzymes in a mostly carbohydrate-depleted medium suggests that most CDZymes are permanently expressed regardless of substrate availability. We further note that in previous cultivation studies, the lysosomal CDZymes identified in this study showed conditional enrichment, while the newly identified CDZymes of the cell membrane do not change in the face of different conditions or media (Fig 1D) [8]. Such constitutive presence supports suggestions recently made for plants and algae being the primary food source of *P. papillatum* in nature, potentially making use of both carbohydrates on the external cell walls, as well as the internal proteinaceous energy sources [9].

The soluble lysosome contains a chitinase (Fig 3C), while in the endocytic trafficking compartment we documented a complementary glucuromannan-digesting GH92, which combined suggests a proclivity for fungal cell wall digestion (Fig 3B). The single observation of *P. papillatum* regarding its *in natura* behavior comes from its initial isolation from

drifting eelgrass [31], a plant that is known to harbor various fungal cohabitants on its surface [32]. This documented enzymatic sub-localization appears consistent with such a supposition for varied sources of nutrition. Interestingly, a single CDZyme member, xylan-α-glucuronidase, is predicted with high confidence to the cytosol, despite the presence of an N-terminal signal sequence. This enzyme is additionally predicted to have been acquired *via* horizontal gene transfer from a bacterial endosymbiont, for which diplonemids have shown a propensity for acquisition [33,34], though absent from the extant *P. papillatum*.

**Subcellular distribution of glycolysis/gluconeogenesis enzymes reveals novel glycosomal insights for *P. papillatum***

Diplonemids and their sister lineage, the mostly parasitic kinetoplastids, are categorized as glycomonads due to their shared compartmentalization of part of their glycolytic pathways in specialized peroxisomes called glycosomes [35], yet the extent to which these organelles retain the same function in both lineages is unclear. Kinetoplastids localize the first seven steps of glycolysis to glycosomes [20], while in diplonemids peroxisomal targeting signals (PTS) are predicted in six enzymatic steps suggesting a similar metabolic arrangement, although only five enzyme have been experimentally confirmed to colocalize with known peroxisomal proteins [36,37]. Here, we use our subcellular proteomics dataset to partly confirm and expand on previous analyses (Fig 4). As our glycosome showed fractional similarity with other organelle clusters, we only confirmed two enzymatic steps to this organelle, one of which (step III) representing a newly described designation in *P. papillatum* (Fig 4A). However, we confirmed the cytosolic localization of four enzymes, glyceraldehyde 3-phosphate dehydrogenase (GADPH, step VI), phosphoglycerate mutase (PGAM, step VIII), enolase (step IX), and pyruvate kinase (Step Xa) [37].

Certain enzymes were not detected in previous investigations; thus, it came as a surprise that we detected in the glycosome both phosphofructokinase (PFK) and fructose 1,6-biphosphatase (FBP), which typically participate in glycolysis (IIIa) and gluconeogenesis (IIIb), respectively (Figs 4A and S5). Their localization demonstrates a capacity for this organelle to mediate both directions of this pathway (Fig 4A). The genome of *P. papillatum* encodes two PFKs [37], with PFK1 (DIPPA_21987) being a PPi-dependent variant horizontally acquired from a bacterium [8], which is typically able to function in an ATP-poor environment. PFK1 also shows the potential to engage in gluconeogenesis [8], which, along with FBP, further supports the capacity of *P. papillatum* "glycosomes" to perform steps of gluconeogenesis. We further note the prediction of a TMD in PFK1, the presence of which represents an unusual feature for enzymes of this pathway (Fig 4A), though not without precedent [38]. We propose that the N-terminal TMD allows insertion of the enzyme from within the glycosome, exposing its enzymatic domains to the organellar lumen (S6A Fig). While previous transcriptome analysis recorded an additional PTS1-lacking PFK with presumable cytosolic residence [8], a survey of the now complete genome confirmed only the presence of PFKs furnished with PTS [9].

One copy (DIPPA_70192) of fructose-biphosphate aldolase (FBA, step IV) was previously localized to the glycosomes and indeed shows a corresponding fractional pattern in our dataset (Figs 4A and S5 Fig). A second FBA (DIPPA_30805), also bearing a PTS2 motif, displays a more subdued profile with less similarity to the cytosolic and glycosomal fractional profiles (S6B Fig). We interpret this as co-localization in both compartments, a phenomenon described for several peroxisomal proteins across eukaryotes [39]. Moreover, while in 'Hemi' media, the PTS1-bearing copies of glucose 6-phosphate isomerase (G6P) and triosephosphate isomerase (TIM) have been localized to the glycosomes [37], in our dataset they occupy an ambiguous position that similarly implies a dual localization, which contrasts with their confidently placed cytosolic counterparts which lack a PTS (Figs 4A, S6C, and S6D).

We additionally localized multiple copies of PTS-lacking GADPH distributed to either the cell membrane or the mitochondrion, as described elsewhere [40]. We demonstrate a convincing cytosolic localization for four additional paralogues of enzymes lacking a PTS, namely G6P (step II), TIM (step V), phosphoglycerate kinase (PGK, step VII), and PGAM (Fig 4A). While such cytosolic localizations may facilitate the glycolytic processing from glyceraldehyde-3-phosphate to

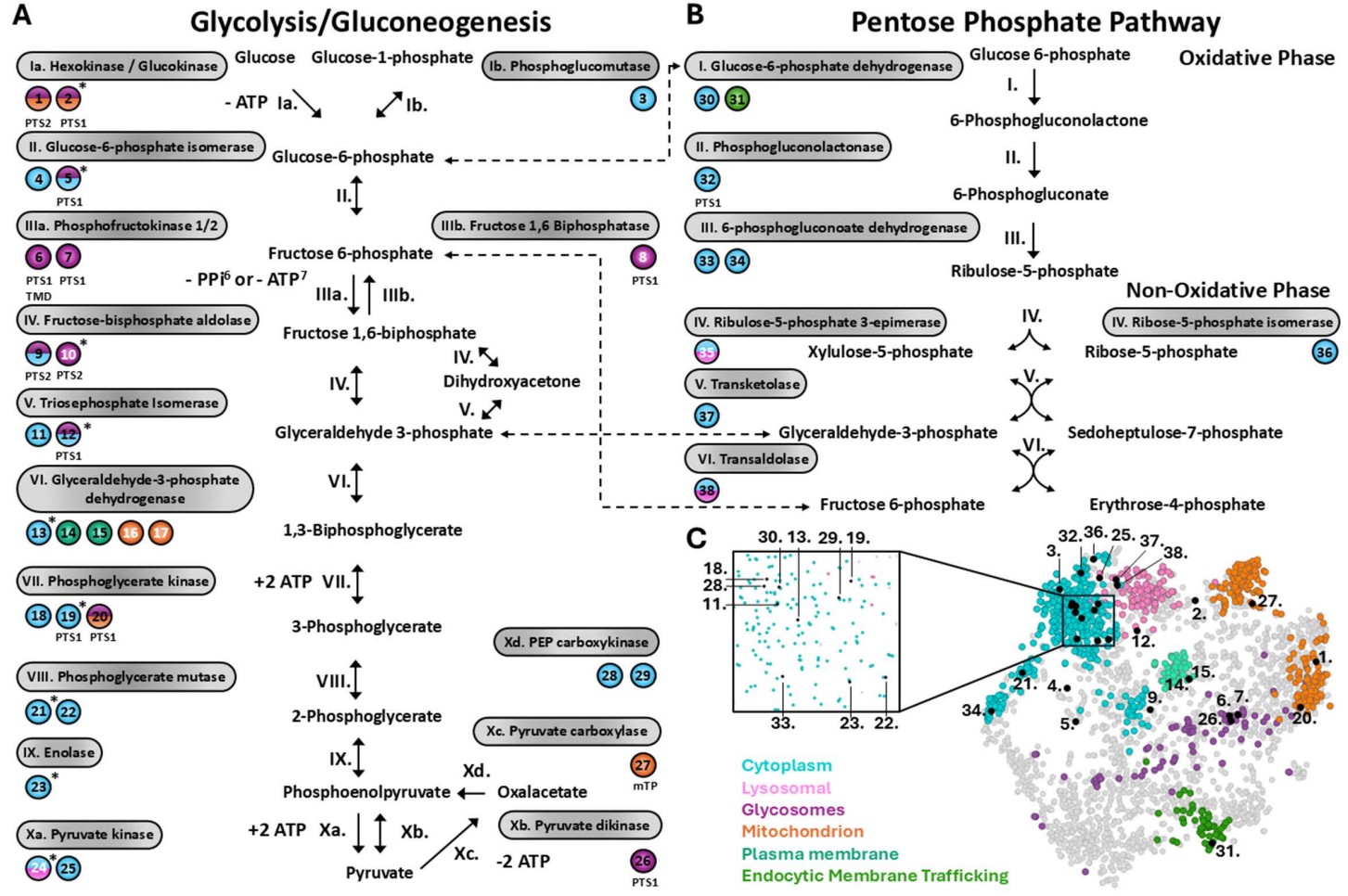

**Fig 4. Metabolic reconstruction of glycolysis/gluconeogenesis and pentose phosphate pathway demonstrates altered glucose metabolism in *Paradiplonema papillatum*.** Localization of relevant enzymes across glycolysis/gluconeogenesis **(A)** and pentose phosphate pathway **(B)**, resolved on neighbor-average imputed t-SNE **(C)** with relevant localization clusters highlighted. Peroxisomal target sequences (PTS), mitochondrial target peptides (mTP), and transmembrane domains (TMD) are indicated. Proteins previously localized via anti-sera immunolocalizations indicated with *, metabolite shunts between two pathways indicated with dotted arrows. Split coloring of proteins represents their manual designations to the cytosol (24,25,38) or indicates the possibility of glycosomal dual localizations between the cytosol and glycosomes (1,2,5,9,12,20), based on inspection of fractionation pro-files (S6 Fig) and targeting signals. Protein numbers highlighted in white represent those only resolved on zero and neighbor-average imputed t-SNE (S5 Fig). Further information is available in Table G in S1 Data.

pyruvate, they alternatively reveal the potential for partial cytosolic gluconeogenesis initiated by processing oxaloacetate to phosphoenolpyruvate *via* cytosol-localized PEP carboxykinase (Fig 4A). Along with other reversible steps of glycolysis, proteomic enrichment was reported for this enzyme from cells grown in the glucose-depleted "Hemi" medium [8], reflecting an increased use of gluconeogenesis under such conditions (Table G in S1 Data).

Metabolically adjacent to glycolysis and gluconeogenesis is the pentose phosphate pathway (PPP), which facilitates the interconversion of simple carbohydrates of different sizes (Fig 4B). In kinetoplastids, several PPP enzymes possess a PTS, producing a glycosomal or dual glycosomal and cytosolic localization [41]. However, *P. papillatum* encodes only a single PTS1-possessing enzyme, phosphogluconolactonase (step II) (Fig 4B). Despite its targeting signal, our data-set suggests the enzyme localizes to the cytoplasm (S6E Fig), demonstrating that, similar to the localizations of certain

proteins in *T. brucei* [42], the presence of a PTS does not guarantee peroxisomal targeting (Fig 4B and 4C). While ribulose-5-phosphate 3-epimerase (step IV of PPP) and *trans*-aldolase (step VI of PPP) are classified to the soluble lysosome, considering the fractional similarity of this cluster to that of the cytosol, we regard them as cytosolic (Figs 4A–4C and S5). By contrast, a copy of glucose-6-phosphate dehydrogenase (step I of PPP) shows a fractionation profile consistent with that of the endocytic membrane trafficking cluster, which warrants future investigation (Figs 4B, 4C, and S6F).

In summary, our localization of individual steps for glycolysis/gluconeogenesis supports the hypothesis that diplonemids separated from kinetoplastids prior to the complete transfer of the first seven steps of these pathways into the glycosomes [37], leaving step VI in the cytosol for diplonemids. Moreover, in these flagellates the cell membrane-embedded versions of GAPDH underlines compartmentalization of this enzyme distinct from that of its homologs in kinetoplastids [43]. A further distinction is represented by PEP carboxykinase, which in *P. papillatum* remains cytosolic and lacks a PTS, with its glycosomal compartmentalization evolved in the kinetoplastid clade only secondarily. The presence of a PTS1 in just a single PPP enzyme likely signifies a remnant of the ancestral trend in glycomonads towards compartmentalization of this pathway which, unlike in kinetoplastids, did not continue to progress in diplonemids.

### Versatile amino acid digestive capabilities within the mitochondrion

Gluconeogenesis in *P. papillatum* is presumably supplied by substrates from the AA catabolism [37], which we endeavored to resolve with our subcellular dataset. Accordingly, we show this protist's capacity to digest a broad range of AA's, primarily within the mitochondrion, in accordance with *in silico* predictions [44] and reminiscent of similar capabilities demonstrated in the mitochondrion of its fellow euglenozoan, *Euglena gracilis* [45] (S7 Fig). Ultimately, the mitochondrion of *P. papillatum* appears capable of metabolizing arginine, aspartate, histidine, glutamate, glycine, isoleucine, leucine, proline, serine, threonine, and valine into metabolites that can directly feed into the tricarboxylic acid (TCA) cycle, as well as glutamine and cysteine with initial cytosolic processing (S7 Fig). We additionally reveal the possibility of diplonemids to process the fatty acid propanoate to propanoyl-CoA, which would allow incorporation into AA intermediate-processing pathways within the mitochondrion, representing a functional distinction from kinetoplastids (S7 Fig).

### Conclusions

Previous work has demonstrated the global distribution, relative importance, abundance, and diversity of marine diplonemids [6], underscoring the value in clarifying their ecological roles and biology. Only recently has *P. papillatum* emerged as a genetically tractable species [12], opening the entire clade to inquiry *via* cellular and molecular methods. Our subcellular proteomics dataset is complementary to these efforts and provides a pathway towards hypothesis-driven research, thereby accelerating our understanding of these ecologically and evolutionary important protists [13,14,16,17]. In total, our data enabled us to localize thousands of proteins to 22 distinct subcellular compartments in *P. papillatum*. The confidence of our data is strengthened by the endogenous tagging of selected proteins.

From this wealth of data, we focused specifically on the confidently predicted cluster of cell membrane proteins. In this cluster, we identified an expanded family of CDZymes, supporting recent predictions that *P. papillatum* primarily feeds on plants and algae *via* degrading their cell walls. CDZymes were also localized to the lysosome, further suggesting active ingestion of complex carbohydrates. The fact that we supplied *P. papillatum* with protein-rich, carbohydrate-limited media represents an intriguing question for future analysis: why are CDZymes expressed in the absence of carbohydrates? We speculate that they are produced in anticipation of interacting with these substrates, and hope that *in natura* studies may now be used to definitively clarify the ecological role for this and other diplonemids.

In conclusion, we have sub-localized thousands of proteins in a model species representing a major protist group. Given the scarcity of available marine protists that are genetically tractable and can be investigated with relative ease [46], our data provide a novel and rich resource to explore diplonemids' unique cell biology and to map ancestral traits in this free-living heterotrophic flagellate.

## Materials and methods

### Resource availability

Experimental resources used in this study are listed in Table 1. The mass spectrometry proteomics data have been deposited to the ProteomeXchange Consortium via the PRIDE partner repository with the dataset identifier PXD065121 [51]. Further information and requests for reagents should be directed to and will be fulfilled by the lead contact, Michael J. Hammond (michael.hammond@paru.cas.cz).

### Materials availability

Vectors and novel cell lines generated for this study are available from lead contact upon request.

### Experimental model and study participant details

*P. papillatum* (ATCC50162) served as the cell line for both proteomic analysis and cell line generation.

### Strain, culture conditions, and preparation for lysis

*Paradiplonema papillatum* (ATCC50162) cells were cultivated axenically in Diplo media (36 g/L sea salts [Sigma], with 1 g Tryptone and 10 ml Fetal Bovine Serum [Sigma], 0.22 µm filter sterilized) at 22°C. Cell cultures were harvested in a combined volume of 750 ml per sample, harvested at ~$2.5 \times 10^6$ cells/ml, and processed in triplicates. Cell cultures were concentrated by centrifugation (900$g$ for 10 min) and pellets were resuspended in 6 ml detergent-free lysis buffer (0.25 M sucrose, 10 mM HEPES, pH 7.4, 2 mM EDTA, 2 mM Mg(OAc) with Halt Protease and Phosphatase Inhibitor Cocktail, pre-chilled to 4°C.

### Cell lysis and fractionation

Cell suspension underwent lysis *via* nitrogen cavitation at 250 psi for 10 min (Parr 4639, Parr Instrument). Cell lysate was gently released from the chamber to minimize foaming, with collected sample undergoing differential centrifugation following a previously established protocol [19]. Briefly, cell lysate underwent centrifugation at speeds (Table 2), and the supernatant was transferred to fresh 2 ml centrifuge tubes and subjected to subsequent centrifugation steps, with pellets from previous spins stored at −80°C after collection, in addition to the supernatant fraction from final spin. Pelleted cell lysate was additionally collected and stored for proteomic analysis.

### Fractional assessment

To assess the distribution and enrichment of proteins across *P. papillatum* fractions, immunoblotting was performed using antibodies against ATP synthase subunit β (kindly provided by A. Zíková) [21], Grp75 (Enzo) [22], and Grp78 (Novus) [23]. Pellets were resuspended in 2× Laemmli buffer (0.125 M Tris-HCl, pH 6.8, 4% SDS, 20% glycerol, 0.004% bromophenol blue) without DTT. Fractional samples were quantified using the Pierce BCA Protein Assay (Thermo Fisher Sci.).

10 µg of protein was loaded onto an SDS-PAGE gel (Invitrogen Bolt Bis-Tris Plus Mini Protein Gels, 4%–12%, 1.0 mm, WedgeWell format) along with a protein marker (Amersham ECL Rainbow Marker—Full Range). The gel was run for 1 hour at 130 V, briefly washed in 1× PBS buffer, and transferred onto a methanol-activated PVDF membrane (iBlot 2 Transfer Stacks, PVDF, Invitrogen) using the iBlot 2 Western Blot Transfer Device (Invitrogen). The membrane was blocked in 5% non-fat dry milk and 1× PBS buffer for 1 hour at room temperature, followed by incubation with relevant antibodies diluted (1:10,000 for ATP-β, and 1:1,000 for Grp75 and Grp78) in blocking solution (5% non-fat dry milk in 1× PBS buffer). Blots were incubated at room temperature for 1 hour, then overnight at 4°C. The following day blots were washed three times for 10 min each in 1× PBS, probed with HRP-linked secondary antibodies (31460/31430, Invitrogen) diluted 1:1,000 in blocking solution for 1 hour at room temperature, and rinsed again three times for 10 min each in 1× PBS-T. Detection

**Table 1. Key resources employed within this study.**

| Reagent or Resource | Source | Identifier |
|---|---|---|
| Antibodies | | |
| Rabbit anti-ATP Synthase-β | Zíková and colleagues [21] | |
| Mouse anti-GRP 75 | Enzo | Cat# SPS-826D, RRID:AB_2120451 (https://www.antibodyregistry.org/AB_2120451) |
| GRP 78 | Novus | Cat# NB100-56413, RRID:AB_838320 (https://www.antibodyregistry.org/AB_838320) |
| Goat anti-Rabbit IgG (H+L) Secondary Antibody, HRP | Invitrogen | Catalog# 31460 (https://www.thermofisher.com/antibody/product/Goat-anti-Rabbit-IgG-H-L-Secondary-Antibody-Polyclonal/31460) |
| Goat anti-Mouse IgG (H+L) Secondary Antibody, HRP | Invitrogen | Catalog# 31430 (https://www.thermofisher.com/antibody/product/Goat-anti-Mouse-IgG-H-L-Secondary-Antibody-Polyclonal/31430) |
| Mouse anti-V5 Monoclonal Antibody (2F11F7) | Invitrogen | Catalog# 37–7500-A555, RRID:AB_2610631 (https://www.antibodyregistry.org/AB_2610631) |
| Rabbit anti-V5 Polyclonal Antibody | Sigma-Aldrich | Catalog# V8137, RRID:AB_261889 (https://www.antibodyregistry.org/AB_261889) |
| Goat anti-Mouse IgG (H+L) Cross-Adsorbed Secondary Antibody, Alexa Fluor 555 | Invitrogen | Catalog# A-21422 (https://www.thermofisher.com/antibody/product/Goat-anti-Mouse-IgG-H-L-Cross-Adsorbed-Secondary-Antibody-Polyclonal/A-21422) |
| Goat anti-Rabbit IgG (H+L) Cross-Adsorbed Secondary Antibody, Alexa Fluor 488 | Invitrogen | Catalog# A-110087 (https://www.thermofisher.com/antibody/product/Goat-anti-Rabbit-IgG-H-L-Cross-Adsorbed-Secondary-Antibody-Polyclonal/A-11008) |
| Chemicals, peptides, and recombinant proteins | | |
| Critical commercial assays | | |
| Pierce Dilution-Free Rapid Gold BCA Protein Assay | Thermo Scientific | Catalog# A55860 |
| Pierce Quantitative Peptide Assays and Standards | Thermo Scientific | Catalog# 23290 |
| Deposited data | | |
| Raw peptide data | PRIDE | PXD065121 |
| For protein predictions and annotations see Table D in S1 Data | | |
| Experimental models: Cell lines | | |
| *Paradiplonema papillatum* | Porter [31] | ATCC50162 |
| Experimental models: Organisms/strains | | |
| For cell lines generated for this study see Table E in S1 Data | | |
| Oligonucleotides | | |
| For primers used in this study see S2 Data. | | |
| Recombinant DNA | | |
| pBA3294 vector | Akiyoshi and colleagues [13] | |
| pDP011 vector | Genebank | OQ547858 |
| Software and algorithms | | |
| Micosoft Excel | Microsoft | https://www.microsoft.com |

*(Continued)*

**Table 1.** (Continued)

| Reagent or Resource | Source | Identifier |
|---|---|---|
| R and Rstudio | Rstudio | https://posit.co/download/rstudio-desktop/ |
| pROLOC | Crook and colleagues [47] | |
| Fiji (Image J) | Fiji | https://fiji.sc/ |
| Signal P 6.0 | Teufel and colleagues [27] | https://services.healthtech.dtu.dk/services/SignalP-6.0/ |
| Target P 2.0 | Armenteros and colleagues [26] | https://services.healthtech.dtu.dk/services/TargetP-2.0/ |
| DeepTMHMM | Hallgren and colleagues [28] | https://services.healthtech.dtu.dk/services/DeepTMHMM-1.0/ |
| DeepLOC 2.1 | Odum and colleagues [48] | https://services.healthtech.dtu.dk/services/DeepLoc-2.1/ |
| NetGPI 1.1 | Gíslason and colleagues [49] | https://services.healthtech.dtu.dk/services/NetGPI-1.1/ |
| Ghost KOALA | Kaneisha and colleagues [50] | https://www.kegg.jp/ghostkoala/ |

**Table 2. Fractional protocol for centrifugation as used in this study, adapted from LOPIT-DC protocol [19].**

| Fraction | Centrifuge speed (x *g*) | Spin time (min) |
|---|---|---|
| Cell Lysate | 200 | 5 |
| 1 | 1,000 | 10 |
| 2 | 3,000 | 10 |
| 3 | 5,000 | 10 |
| 4 | 9,000 | 15 |
| 5 | 12,000 | 15 |
| 6 | 15,000 | 15 |
| 7 | 30,000 | 20 |
| 8 | 79,000 | 43 |
| 9 | 120,000 | 45 |
| Supernatant | NA | NA |

was performed using the Pierce ECL Western Blotting Substrate (Thermo Fisher Sci.), and imaging was conducted with the Azure 600 (Biosystems).

## Sample preparation and LC–MS Analysis

Native protein pellets obtained from differential centrifugation were digested and desalted following the protocol for the S-Trap Micro Column (ProtiFi, USA). Protein concentration was quantified using the BCA assay (Thermo Fisher Sci.), while peptide concentration was measured using a fluorometric kit (Thermo Fisher Sci.).

## Liquid-chromatography tandem mass spectrometry

LC–MS/MS analyses were performed at the Biosciences Mass Spectrometry Core Facility, Arizona State University. Data-dependent mass spectra were collected in positive mode using an Orbitrap Fusion Lumos mass spectrometer coupled with an UltiMate 3000 UHPLC (Thermo Fisher Sci.). Peptides were fractionated on an Easy-Spray LC column

(50 cm × 75 μm ID, PepMap C18, 2 μm, 100 Å) with an upstream trap column. Each sample was analyzed in technical triplicate. LC–MS settings: electrospray potential 1.6 kV, ion transfer tube temperature 300°C, and the "Universal" peptide analysis method. Full MS scans (375–1,500 $m/z$) were acquired at a resolution of 120,000 with 3 s cycles. The RF lens was set to 30%, AGC to "Standard," and monoisotopic peak determination included charge states 2–7. Dynamic exclusion was 60 s with a 10 ppm mass tolerance. MS/MS spectra were acquired in centroid mode with a quadrupole isolation window of 1.6 $m/z$ and CID energy of 35%. Peptides were eluted over a 240-min gradient at 0.25 μL/min using 2%–80% acetonitrile/water: 0–3 min (2%), 3–75 min (2%–15%), 75–180 min (15%–30%), 180–220 min (30%–35%), 220–225 min (35%–80%), 225–240 min (80%–85%).

LC–MS/MS analysis of the digested peptides was performed on an EASY-nLC 1200 (Thermo Fisher Sci.) coupled to an Orbitrap Eclipse Tribrid mass spectrometer (Thermo Fisher Sci.). Peptides were separated on an Aurora UHPLC column (25 cm × 75, 1.6 μm C18, AUR2-25075C18A, Ion Opticks) with a flow rate of 0.35 μL/min for a total duration of 135 min ionized at 1.6 kV in the positive ion mode. The gradient was composed of 2% solvent B (5 min), 2%–6% B (7.5 min), 6%–25% B (82.5 min), 25%–40% B (30 min), 40%–98% B (1 min) and 98% B (15 min); solvent A: 2% ACN and 0.2% FA in water; solvent B: 80% ACN and 0.2% FA. MS1 scans were acquired at the resolution of 120,000 from 350 to 1,600 $m/z$, AGC target 1e6, and maximum injection time 50 ms. MS2 scans were acquired in the ion trap using fast scan rate on precursors with 2–7 charge states and quadrupole isolation mode (isolation window: 0.7 $m/z$) with higher-energy collisional dissociation (HCD, 30%) activation type. Dynamic exclusion was set to 30 s. The temperature of ion transfer tube was 300°C and the S-lens RF level was set to 30.

### Raw data processing and quantification

The LFQ analysis was performed using Proteome Discoverer 2.4 (Thermo Fisher Sci.) based on the composite database: *P. papillatum's* predicted proteome, and mitochondrial ORFs, Raw files were searched with SequestHT using Trypsin as the enzyme, allowing up to three missed cleavages. Peptide length was set to 6–144 AAs, with precursor ion mass tolerance at 20 ppm, fragment mass tolerance at 0.5 Da, and a minimum of one peptide identified. Carbamidomethyl (C) was a fixed modification, while Acetyl (N-terminus), Met-loss (N-terminus), and oxidation of Met were dynamic modifications. A target/decoy strategy and 1.0% FDR were calculated using Percolator. Data were imported into Proteome Discoverer 2.4, and features were detected using the Minora Feature Detector algorithm. The area under the curve for aligned ion chromatograms was calculated to determine relative abundances.

Proteins and their corresponding LFQ abundance values were imported into the R programming language and converted into MSnset object using the Bioconductor packages MSnbase (v 2.24.2) and pRoloc (v 1.38.2) [47]. The data was examined and proteins with low confidence (PSM < 3 and without unique peptides) were filtered out. Triplicates were averaged to generate a 33rd-dimensional dataset of relative protein abundance. The datasets were split into their respective experiments (i.e., 1–11, 12–22, 23–33) to perform hybrid imputation and sum-normalization across rows.

Missing data were imputed first by nearest-neighbor averaging and then imputing zeros for all remaining empty cells. Principal component analysis and t-SNE were applied for dimensional reduction and data visualization.

### Supervised and unsupervised classification

268 manually curated marker proteins (Table B in S1 Data) were used as the training set for a support vector machine (SVM) model with the "svmOptimization" and "svmClassification" functions in pRoloc package. Initially, 100 rounds of 5-fold cross-validation were performed to optimize the SVM parameters based on the marker protein abundance profiles. The optimal parameters for the SVM classifier were then applied to all proteins in the dataset with a corresponding SVM score whose range is 0–1 with 1 being the score of marker proteins. The SVM classifier was then applied to unlabeled data (i.e., non-marker proteins) with corresponding weights applied to each marker class. Each protein was thus classified

to one compartment, and any protein whose classification fell below the global median SVM score was reset to "unknown" while the other half of the dataset was considered "predicted" to its corresponding compartment due to their higher SVM scores (Table D in S1 Data).

Unsupervised clustering was performed using the K-means (KM) algorithm implemented in the MLearn function from the MLInterfaces package in Rstudio (version 1.78.0). KM generates k random centroids and includes surrounding data points iteratively such that all data points are included in one of the k clusters and the size of each centroid is minimized. KM clusters were generated with 22 clusters corresponding to number Rof marker groups (Table D in S1 Data).

**Targeting signal prediction, annotation, and conditional enrichment analysis**

*P. papillatum* protein database was annotated via blast search against CDS of parasitic kinetoplastid *Trypanosoma brucei* 927 (v66) and free-living *Bodo saltans* (v66) (https://tritrypdb.org/tritrypdb/app) as well as baker's yeast *Saccharomyces cerevisiae* (559292) (https://blast.ncbi.nlm.nih.gov/Blast.cgi), with a threshold of $E^{-5}$. Metabolic pathway analysis was also performed via GhostKoala [50].

Signal P version 6.0 was used for the prediction of signal peptides, using a confidence threshold of >0.9 (Fig 1C) [27], with NetGPI 1.1 additionally used on this subset to determine proteins that additionally possessed predicted C-terminal GPI anchors [49] (Table D in S1 Data). Target P 2.0 was used for prediction of mitochondrial target peptides [26], with DeepTMHMM [28] used for predictions of TMD (Fig 1C) (Table D in S1 Data). Peroxisomal target signal prediction was conducted using a custom regex script designed by Prof. Fred Oppoerdoes against a broad range of AA combinations with PTS1 determined by the script: [SAGCNP][RHKSNQ][LIVFAMY]\$, and PTS2 via ^M.[1,10],[RK][LVI].....[HQ][ILA] (Table D in S1 Data), which were then manually inspected for specific enzymes of relevance (Tables F and G in S1 Data). DeepLoc2.1 was additionally used to assess protein localization predictions and membranous status [48] (Table D in S1 Data).

Protein enrichment data for media and conditional cultivation [8] was displayed across dataset, including proteins that displayed enrichment status of any capacity (Fig 1D) (Table D in S1 Data).

**Endogenous tagging and *P. papillatum* microscopy**

Endogenous C-terminal tagging of cell lines corresponding to 12 proteins within supervised protein clusters were generated to verify predictions (Table E in S1 Data).

Proteins DIPPA_11651.mRNA.1, DIPPA_15120.mRNA.1, DIPPA_04811.mRNA.1, DIPPA_32825.mRNA.1, DIPPA_00315.mRNA.1 underwent tagging via yellow fluorescent protein, using vector pBA3294 [13]. *Pac*I and *Asc*I restriction sites of pBA3294 were used to insert two ~2 kb homology arms that were amplified from genomic DNA by PCR using KOD one polymerase (Merck). Primer sequences are provided in S2 Data. The first fragment corresponds to downstream of the gene ORF (starting just after its stop codon) surrounded with *Pac*I and *Not*I restriction sites, while the second fragment corresponds to the 2 kb DNA fragment starting from 2kb upstream of the stop codon and ending just before the stop codon surrounded with *Not*I and *Asc*I. After cutting the fragments with respective restriction enzymes, the two DNA fragments were ligated into pBA3294 that were cut with *Pac*I and *Asc*I. Plasmids were validated by nanopore whole plasmid sequencing (Plasmidsaurus). Tagging constructs were linearized by *Not*I, transfected into *P. papillatum* cells by electroporation, and selected by the addition of 75 µg/mL G418.

Cells were pelleted by centrifugation at 1,300*g* for 5 min and fixed by 4% formaldehyde solution diluted in PBS for 5 min. Cells were washed with 1 mL PBS twice, resuspended in a small volume of DABCO mounting media (1% w/v 1,4-diazabicyclo[2.2.2]octane, 90% glycerol, 50 mM sodium phosphate pH 8.0) with 100 ng/mL DAPI, and mounted onto glass slides. Images were captured on an Axioimager.Z2 microscope (Zeiss) installed with ZEN using a Hamamatsu ORCA-Flash4.0 camera with 63× objective lenses (1.40 NA). Typically, 25 z sections spaced 0.24 µm apart were collected.

Proteins DIPPA_07493.mRNA.1, DIPPA_20982.mRNA.1, DIPPA_24150.mRNA.1, DIPPA_16310.mRNA.1, DIPPA_24837.mRNA.1, DIPPA_21158.mRNA.1, DIPPA_16504.mRNA.1 underwent tagging via 3×V5 epitope, using vector pDP011 (GeneBank OQ547858) [52] (Table E in S1 Data). A fusion PCR strategy using Q5 High-Fidelity DNA Polymerase (NEB Biolabs, M0491S) was used to design and obtain the above DNA constructs, as described previously [12]. Used primers and product sizes are listed in S2 Data. 1–5 µg of gel-purified and ethanol-precipitated DNA constructs were electroporated into $5 \times 10^7$ cells/ml *P. papillatum* cells as described elsewhere [12,53]. Twenty-four h after electroporation, transfected cells underwent selection in a 24-well plate at 27°C, under increasing concentrations of hygromycin (100–225 µg/mL). After 3 weeks, transfectants were selected and expanded into a volume of 10 ml before downstream analyses.

To address subcellular localization of the tagged proteins, an immunofluorescence assay was performed as described previously [52]. Briefly, 20–30 ml of a log phase culture was harvested by centrifugation at 1,700*g* for 10 min, resuspended in 500 µl of 4% paraformaldehyde (dissolved in sea water), and fixed for 15 min on Superfrost plus slides (Thermo Fisher Sci.) at room temperature. After removing the fixative with 1× PBS, cells were permeabilized in ice-cold methanol for 10 min and rinsed with 1× PBS. From this point on, the slides were kept in a humid chamber. Next, the slides were blocked in 5.5% (w/v) fetal bovine serum in PBS-T for 45 min at room temperature, and the blocking solution was removed by washing the cells two times with 1× PBS. The slides were incubated with either mouse anti-V5 or rabbit anti-V5 primary antibody diluted (1:500; Thermo Fisher Sci.) in 3% (w/v) bovine serum albumin (Sigma), at 4°C overnight, covered with parafilm. Afterwards, the primary antibody was removed by washing the slides three times with PBS-T and twice with 1× PBS. AlexaFluor555-labeled goat anti-mouse (1:1,000; Invitrogen) or AlexaFluor488-labeled goat anti-rabbit (1:1,000; Invitrogen) secondary antibody was added and incubated at room temperature for 1 hour in the dark, covered with parafilm. After that, the slides were rinsed three times with PBS-T and twice with 1× PBS. All slides were coated with ProLong Gold Antifade Mountant with DNA Stain DAPI (Life Technol.) and mounted. Samples were imaged with an Olympus BX63 automated fluorescence microscope equipped with an Olympus DP74 digital camera. Pictures were acquired with the cellSens Dimension software (Olympus) and processed through the ImageJ software.

## Supporting information

**S1 Fig. Immunoblot analysis used to resolve fractional distribution across triplicate samples.** 10 µg of protein has been loaded for each fraction generated via differential centrifugation in addition to the initial cell lysate (CL). ATP synthase-β antibody used at 1:10,000 ratio **(A)**, Grp75 antibody used in 1:1,000 **(B)**, and Grp78 **(C)** which displays non-specific signal. Unlysed cells (UC), Supernatant (S). Marker band molecular weights (kDa) indicated in dark gray on the leftmost lane of blots.
(TIF)

**S2 Fig. Neighbor-averaged and zero imputed t-SNE of clustered protein predictions, protein features, and conditional enrichment of dataset. (A)** Full dataset displaying clustered predictions displayed for 4,780 proteins across 22 cell compartments. Predictions were generated via support vector modeling conducted on fractional profiles of marker proteins, applied to the remaining dataset. **(B)** Selected fractional abundances of marker proteins across one replicate of this experiment, representing distinct profiles that facilitate predictive clustering (SUP, Supernatant). **(C)** Software prediction for protein features of signal peptides, transmembrane domains, and mitochondrial target peptides across dataset, demonstrating accumulation across certain defined compartments. **(D)** Proteins determined to be enriched in varying nutrient media (Diplo or Hemi) or cultivation conditions (aerobic or anaerobic) from a conditional study of *Paradiplonema papillatum* [8]. Additional information for all proteins available in Tables B, C and D in S1 Data.
(TIF)

**S3 Fig. Neighbor-averaged and zero imputed t-SNE of endogenous tagged cell lines.** Tagged proteins highlighted (black) among relevant predicted clusters, resolved on neighbor-averaged and zero imputed t-SNE. Individual cell lines were generated via endogenous tagging and imaged through fluorescence microscopy for comparison with compartment relevant protein was predicted to. In descending order, panels depict phase contrast, epitope signal (green), nuclear and mitochondrial DNA (blue), with merges below additionally displaying cell membrane outlines traces for all images, excepting L, which shows only trace of the papilla, which lacks epitope signal. All imaged cells are oriented with their apical regions facing right and posterior facing left. Scale bar represents 5 μm. Further information on cell lines is available in Table E in S1 Data.
(TIF)

**S4 Fig. Cell membrane cluster shows enrichment of proteins possessing both predicted signal peptide (SP) glycosylphosphatidylinositol (GPI) anchors.** t-SNE imputed via neighbor-averaging (A) as well as zeroed dataset (B). Signal peptides predicted via Signal P 6.0 with a confidence threshold greater than 0.9, in tandem with NetGPI 1.1 used for GPI predictions. Further information is available in Table D in S1 Data.
(TIF)

**S5 Fig. Metabolic reconstruction of glycolysis/gluconeogenesis and pentose phosphate pathway on neighbor-averaged and zero imputed t-SNE.** Localization of relevant enzymes across glycolysis/gluconeogenesis (A) and pentose phosphate pathway (B), resolved on neighbor-average and zero imputed t-SNE (C) with relevant localization clusters highlighted. Peroxisomal target sequences (PTS), mitochondrial target peptides (mTP), and transmembrane domains (TMD) are indicated. Proteins previously localized via anti-sera immunolocalizations indicated with *, metabolite shunts between two pathways indicated with dotted arrows. Split coloring of proteins represents their manual designations to the cytosol (24,25,38) or indicates the possibility of glycosomal dual localizations between the cytosol and glycosomes (1,2,5,9,12,20), based on inspection of fractionation profiles (S6 Fig) and targeting signals. Further information is available in Table G in S1 Data.
(TIF)

**S6 Fig. Fractional and schematic analysis of specific enzymes mediating carbohydrate metabolism.** Schematic depiction of DIPPA_21987, phosphofructokinase 1 showing phosphofructokinase (PFK) domains, transmembrane domain (TMD), and Peroxisomal Target Signal along with fractional analysis (A), along with fractional profiles of relevant enzymes across glycolysis/gluconeogenesis (B) compared to marker proteins of the cytosol, glycosomes, and endocytic membrane trafficking markers.
(TIF)

**S7 Fig. Metabolic reconstruction of Amino Acid (AAs) breakdown for incorporation in the TCA cycle, localized across cell compartments.** AAs and metabolites of the TCA cycle are indicated in bold. Propanoate metabolism, which involves intermediates of certain AA digestion, is also depicted. Split coloring indicates manual annotation for specific enzymes based on certain target peptides or candidate function, on top, versus contrasting predictions below (e.g., Enzyme 2: proline dehydrogenase, we designate to the mitochondrion, despite low confidence predictions to the nucleus). Further information is available in Table H in S1 Data.
(TIF)

**S1 Data. Tables A–H.**
(XLSX)

**S2 Data. Primer sequences used for endogenous tagging of *Paradiplonema papillatum*.**
(XLSX)

**S1 Raw Images. Immmunoblot images unadjusted and uncropped, used for S1 Fig.**
(PDF)

## Acknowledgments

We thank A. Zíková (Biology Centre) for the anti-ATP synthase subunit β antibodies.

## Author contributions

**Conceptualization:** Michael J. Hammond, Julius Lukes, Jeremy G. Wideman.

**Data curation:** Michael J. Hammond, Yu-Ping Poh.

**Formal analysis:** Michael J. Hammond, Jeremy G. Wideman.

**Funding acquisition:** Bungo Akiyoshi, Julius Lukes, Jeremy G. Wideman.

**Investigation:** Michael J. Hammond, Orsola Iorillo, Drahomíra Faktorová, Bungo Akiyoshi, Jeremy G. Wideman.

**Methodology:** Michael J. Hammond, Orsola Iorillo, Drahomíra Faktorová, Michaela Svobodová, Bungo Akiyoshi, Jeremy G. Wideman.

**Project administration:** Michael J. Hammond, Drahomíra Faktorová, Julius Lukes, Jeremy G. Wideman.

**Software:** Michael J. Hammond, Orsola Iorillo, Tim Licknack, Yu-Ping Poh, Jeremy G. Wideman.

**Supervision:** Drahomíra Faktorová, Julius Lukes, Jeremy G. Wideman.

**Validation:** Michael J. Hammond.

**Visualization:** Michael J. Hammond.

**Writing – original draft:** Michael J. Hammond, Julius Lukes, Jeremy G. Wideman.

**Writing – review & editing:** Michael J. Hammond, Julius Lukes, Jeremy G. Wideman.

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
