## [Editor Report · Decision Letter 0]

14 Jul 2025

Dear Dr Hammond,

Thank you for submitting your manuscript entitled "Subcellular proteomics of Paradiplonema papillatum reveals digestive capacity of the cell membrane and the plasticity of peroxisomes across euglenozoans" for consideration as a Research Article by PLOS Biology.

Your manuscript has now been evaluated by the PLOS Biology editorial staff as well as by an academic editor with relevant expertise and I am writing to let you know that we would like to send your submission out for external peer review.

Once your full submission is complete, your paper will undergo a series of checks in preparation for peer review. After your manuscript has passed the checks it will be sent out for review. To provide the metadata for your submission, please Login to Editorial Manager (https://www.editorialmanager.com/pbiology) within two working days, i.e. by Jul 16 2025 11:59PM.

Kind regards,

Ines

--

Ines Alvarez-Garcia, PhD

Senior Editor

PLOS Biology

---

## [Decision Letter · Decision Letter 1]

10 Oct 2025

Dear Dr Hammond,

Thank you for your patience while your manuscript entitled "Subcellular proteomics of Paradiplonema papillatum reveals digestive capacity of the cell membrane and the plasticity of peroxisomes across euglenozoans" was peer-reviewed at PLOS Biology. Please also accept again my sincere apologies for the delay in sending you our decision. The manuscript has now been evaluated by the PLOS Biology editors, an Academic Editor with relevant expertise, and by two independent reviewers.

The reviews are attached below. As you will see, the reviewers find the conclusions interesting and worth pursuing for publication, however they also raise a few points that would need to be addressed. Reviewer 1 notes a term that is not used properly and should be corrected, whereas Reviewer 2 thinks that most of the enzymes seem to be glycoside hydrolases, rather than CAZymes and recommends changing the term throughout the paper perhaps to carbohydrate degrading enzymes.

Based on the reviews, we are likely to accept this manuscript for publication, provided you satisfactorily address the points raised by the reviewers. However, we would like you to resubmit the manuscript as a Methods and Resources manuscript, thus please select that type of article from the drop-down menu when you resubmit a revised version. Please also make sure to address the data and other policy-related requests stated below my signature.

In addition, we would like you to consider a suggestion to improve the title:

"Subcellular proteomics of the protist Paradiplonema papillatum reveals the digestive capacity of the cell membrane and the plasticity of peroxisomes across euglenozoans"

We expect to receive your revised manuscript within two weeks.

*Published Peer Review History*

*Press*

Sincerely,

Ines

--

Ines Alvarez-Garcia, PhD

Senior Editor

PLOS Biology

Fig. 1B; Fig. S2B and Fig. 6A-F

***Please also make sure you make the files deposited at ProteomeXchange with identifier PXD065121 publicly available at this stage.

CODE POLICY

We require the original, uncropped and minimally adjusted images supporting all blot and gel results reported in an article's figures or Supporting Information files. We will require these files before a manuscript can be accepted so please prepare and upload them now. Please carefully read our guidelines for how to prepare and upload this data: https://journals.plos.org/plosbiology/s/figures#loc-blot-and-gel-reporting-requirements

Reviewers' comments

Rev. 1:

The authors performed sub-cellular fractionation via differential centrifugation followed by proteomics on P. papillatum to identify proteins enriched in different cellular compartments. They did a seemingly thorough job, tested multiple medias and growth conditions and followed up on a few different groups of proteins of interest in their dataset. After manual annotation of 368 proteins and prediction of sub cellular localization for 2,435 they then tested their predictions and confirmed their accuracy by native protein tagging. From this analysis they also identified some interesting protein localizations that will likely fuel a lot of future research and be of interest to the broader community.

The set of CASzymes showed an enrichment of signal-peptide containing proteins, leading the authors to hypothesize a mechanism of extracellular digestion and uptake of nutrients via the cytopharynx. Additionally, they make other predictions about CASzymes and sugar transporters function in various cellular compartments. They also interpret that these CASzymes are constitutively expressed under all of the conditions tested which supports the hypothesis that plants and algae are a primary food source of P. papillatum. Next the authors explored a set of proteins related to the glycolytic pathway and the pentose phosphate pathway. Most of their findings seem to agree with previous findings and/or support existing hypotheses. They did identify some proteins that were localized to compartments that had not been previously described.

Overall their findings shed new light on the evolutionary similarities and differences in comparison to kinetoplastids and reveal a trove of information about the metabolic pathways in a model diplonemid.

As stated above, this work provides independent support for many existing hypothesis as well as providing new predictions about some of the metabolic pathways that exist in diplonemids. The work presented here seems highly impactful to the field as well as the broader community. This work also appears to be technically sound. I would recommend this work for publication in PLoS Biology.

Major comments:

No major comments to be addressed.

Minor Comments:

Most of the language throughout the paper is careful to discuss predictions, but there is at least one place where the language veers more toward demonstration of function, which should be avoided in the absence of functional biochemical studies. (Line 331)

Rev. 2:

I really don't think the CAZymes you mention throughout is the correct term. These nearly all appear to be Glycoside Hydrolases (GHs) and not including glycosyl transferases, auxiliary activities or binding molecules (CBMS) which are also CAZymes. This a bit implies there could be extensive carbohydrate synthesis on the surface if there were lots of GTs. I would prefer this be changed to Glycoside Hydrolases (GH) throughout. Possibly carbohydrate degrading enzymes.

Figure 3A implies there are multiple enzyme domains in eg DIPPA17995 - are these all intact whole enzymes stitched together? Or parts of the active domain split up with spacers between? Were there any carbohydrate binding modules on any of these? Or protein-protein interaction domains to make something akin to a cellulosome? Could the authors make some comment as to this as it is unusual enzyme topology.

Not sure this shows P. papillatum "preys on plants and algae" (line 350), as they could be scavenging shed cell wall or dead cells. Please remove reference to predator/prey and just say feeds upon.

Line 116: "underwent and imputation" Is this meant to be "underwent imputation" or is it "underwent XXX and imputation"?

---

## [Editor Report · Decision Letter 2]

15 Nov 2025

Dear Dr Hammond,

Thank you for the submission of your revised Methods and Resources entitled "Subcellular proteomics of the protist Paradiplonema papillatum reveals the digestive capacity of the cell membrane and the plasticity of peroxisomes across euglenozoans " for publication in PLOS Biology. On behalf of my colleagues and the Academic Editor, Holly Bik, I am delighted to let you know that we can in principle accept your manuscript for publication, provided you address any remaining formatting and reporting issues. These will be detailed in an email you should receive within 2-3 business days from our colleagues in the journal operations team; no action is required from you until then. Please note that we will not be able to formally accept your manuscript and schedule it for publication until you have completed any requested changes.

PRESS

Sincerely,

Ines

--

Ines Alvarez-Garcia, PhD

Senior Editor

PLOS Biology
